



# Combining wake redirection and derating strategies in a wind farm load-constrained power maximization

Alessandro Croce[1], Stefano Cacciola[1], and Federico Isella[1]

[1]Department of Aerospace Science and Technology, Politecnico di Milano, Milano, Italy

**Correspondence:** Alessandro Croce (alessandro.croce@polimi.it)

**Abstract.** Power derating and wake redirection are two wind farm control techniques proposed in the last decade as means for increasing the overall wind farm power output. While derating operations are associated with a limited gain in terms of farm energy harvesting and with a decrease in turbine loading levels, farm controls based on wake redirection proved, both in silico and experimental tests, to entail significant increases in the overall wind farm power output. However, according to wake redirection strategies, the upstream wind turbines may typically operate at large yaw misalignment angles, and the possible increase in loads that the machines may experience in such conditions represents a source of concern when it comes to testing this control on existing farms that are not specifically designed for prolonged misaligned operations. In this work, it is first demonstrated that a suitable derating level can compensate for the increase in the rotor loads associated with large misalignment angles. Secondarily, two load-constrained wind farm controls based on a combination of wake redirection and derating are proposed with the aim of maximizing the overall farm output while maintaining unaltered the design load envelope of the wind turbines operating within the controlled wind farm.

## 1 Introduction

Maximizing the power harvested by wind energy systems and minimizing the associated cost of the energy represent two of the most important goals in the development of any industrial wind energy project. In the last decade, the optimization paradigm has moved from turbine (Bottasso et al., 2022) to farm level (Gebraad et al., 2017, 2016). The concept of wind farm control refers to the synergistic control of all wind turbines within a farm with the goal of maximizing the overall power output, as opposed to the maximization of the single-machine output.

Among all possible techniques that have been proposed as wind farm controls, e.g. wake redirection (Fleming et al., 2019), steady axial induction (Annoni et al., 2016) or dynamic induction control (Frederik et al., 2020; Croce et al., 2023), the one that proved to be highly effective as means of increasing the wind farm energy harvesting is the wake redirection (WR). According to WR technique, the upstream turbine is intentionally yawed so as to deflect its wake out of downstream rotors. Clearly, operations at large yaw misalignment may be detrimental in terms of the loading exerted on the turbine.

As reported in a recent review paper related to the flow control applied to wind farm optimization (Meyers et al., 2022), quantifying the impacts of wind farm control on turbine structural loads represents a critical area of investigation.





In fact, modern wind turbines are designed with the aim of minimizing the associated cost of energy by looking for a good balance between aerodynamic performance and reliability of the structural components (Bortolotti et al., 2016). The international Standards, e.g. IEC 61400-1 Ed.3. (2004); Services (2004), provide the guidelines for the design including a list of design load cases (DLC) to be considered for quantifying fatigue and ultimate loads, as well as maximum structural deflection. An already existing turbine could have been designed without considering the possible impact of wind farm control

on design loads. Consequently, in the case of WR control, the loads and displacements arising from prolonged yawed operations may be a source of concern when it comes to testing or applying a wind farm control technique to already existing farms.

    As argued in Boorsma (2012); Damiani et al. (2018); Croce et al. (2022), operating at large yaw misalignment angles can entail increased design loads, with severity depending on several parameters, such as wind velocity, turbulence intensity, and shear layer.

To cope with this issue, wind farm control definitions employed in real farms often considered strong limitations in the allowable misalignment angles. For example, in a testing campaign reported in Howland et al. (2019) the misalignment turbines were persistently yawed by $20\,\mathrm{deg}$ clockwise for wind coming from a specific sector. In another field campaign, Fleming et al. (2019) employed the so-called "one-sided wake redirection", where only clockwise yaw offsets are allowed. The reason for such a technique lies in the fact that, according to a preliminary study on misalignment-induced loads (Damiani et al., 2018),

clockwise yaw misalignment angles are associated with a lower impact in terms of loads.

    A notable exception to the persistent application of the "one-sided wake redirection" technique to real wind farms is represented by the testing campaign presented in Doekemeijer et al. (2021), in which both negative and positive misalignment angles were employed. However, the maximum misalignment assigned to turbines was limited to $20\,\mathrm{deg}$, and the Authors acknowledged that loads were not considered even if they may play a prominent role.

In this paper, we consider the combination of wake redirection and steady derating controls, that are integrated in a load-constrained wind farm control. Meyers et al. (2022) acknowledged that while wake redirection can be associated with an increase in the loading status of the turbine, derated operations may entail load reduction.

    Clearly, the combination of the aforementioned strategies is not new. Bossanyi (2018) and Debusscher et al. (2022) proposed combined wind farm control with the aim of balancing power harvesting and fatigue loads at wind farm level. Both contribu-

tions suggested that, while wake redirection is exploited for power maximization, derated operations can be optimized so as to achieve a two-fold goal: decreasing fatigue of the turbines and adjusting overall power output to match power demand.

    In this paper, we follow a different methodology that is mainly based on the use of a suitable derating level to compensate for the possible increase in loads induced by yawed operations.

    In particular, the methodology considers three steps. At first the maximum design loads and blade tip deflections, not only

fatigue, are computed for different combinations of the derating level and misalignment angles. In the preliminary investigation, performed in this paper, only the isolated turbine is considered.

    Next, from the map computed in the previous step, the derating level that compensates for the increase in design indicators induced by misalignment is computed. Such a compensating derating level is likely a nonlinear function of the misalignment





angle and can be viewed as a load constraint, that defines a **safe envelope region** in which combined operations, derated and

misaligned, do not lead to increase design loads and blade deflections.

Finally, two open-loop combined and load-constrained controls are proposed by exploiting the safe envelope.

The first is an optimal combined approach in which the derating levels and the misalignment angles of all turbines within the farm are computed so as to optimize the farm power subject to the constraint that the turbine operations be inside the safe envelope.

The second one is a sub-optimal combined control, that only optimizes the misalignment angles without considering load limitations. Afterward, the derating for all turbines is computed by projecting the obtained misalignment on the loads-constraint function.

The controls were tested in a simulation environment, through `Floris` simulations (NREL, 2019) of two-turbine and nine-turbine wind farms.

Both approaches proved effective in increasing the overall farm production, featuring power gains mildly lower than the ones associated with standard unconstrained wake redirection methodologies. Moreover, thanks to the proposed open-loop controls, it is not necessary to impose strong limitations on yaw misalignment angles, such as one-sided policies, as the load constraint automatically ensures load protection.

Moreover, the load-constrained combination of wake redirection and derating, as it is formulated in this work, can be easily

employed in both open- and closed-loop wind farm control algorithms, even if here only open-loop controls are considered.

The paper is divided into three sections. Firstly, Sec. 2 is devoted to the description of the process used for defining the safe envelope and to the formulation of the load-constrained combined control. Secondarily, Sec. 3 deals with the results of all analyses performed to evaluate the load constraint of a 10MW reference wind turbine and to quantify the performance of the proposed controls. Finally, Sec. 4 concludes the paper by summarizing the main findings of the work.

**2   Methodology**

In this section, the methodology followed to define the load-constrained wind farm control combining wake redirection and turbine derating will be detailed.

Such a process considers three consecutive steps. At first, in Sec. 2.1, a parametric analysis is performed in order to quantify the impact of the combination of yaw misalignment and derating operation on the different design loads of a wind turbine.

Secondarily, from the output of the previous analysis, one has to identify the **safe envelope region**, i.e. the region comprising all those combinations of yaw misalignment and derating which are not associated with an increase in the design loads with respect to the reference condition with null misalignment and null derating. Finally, a load-constrained control can be formulated with the goal of finding the optimal wind farm output only within the safe-envelope region.

Figure 1 illustrates this concept in a qualitative way. At first, left plot, one evaluates the increase (or decrease) of different

indicators (such as ultimate and fatigue loads as well as blade deformations) due to derated or yawed operations. Then, middle plot, for all considered indicators, the combinations of derating and yaw misalignment associated with null impact are obtained,





generating multiple constraint lines. At last, right plot, all constraints are merged together to define the safe envelope region, i.e. the region in the plane derating–versus–misalignment that satisfies all the constraints.

Clearly, in order to keep any turbine of the wind farm within its safe envelope, a wind farm control should seek the maximum
power output within the safe envelope of each turbine belonging to the farm, combining this way derated and misaligned operations.

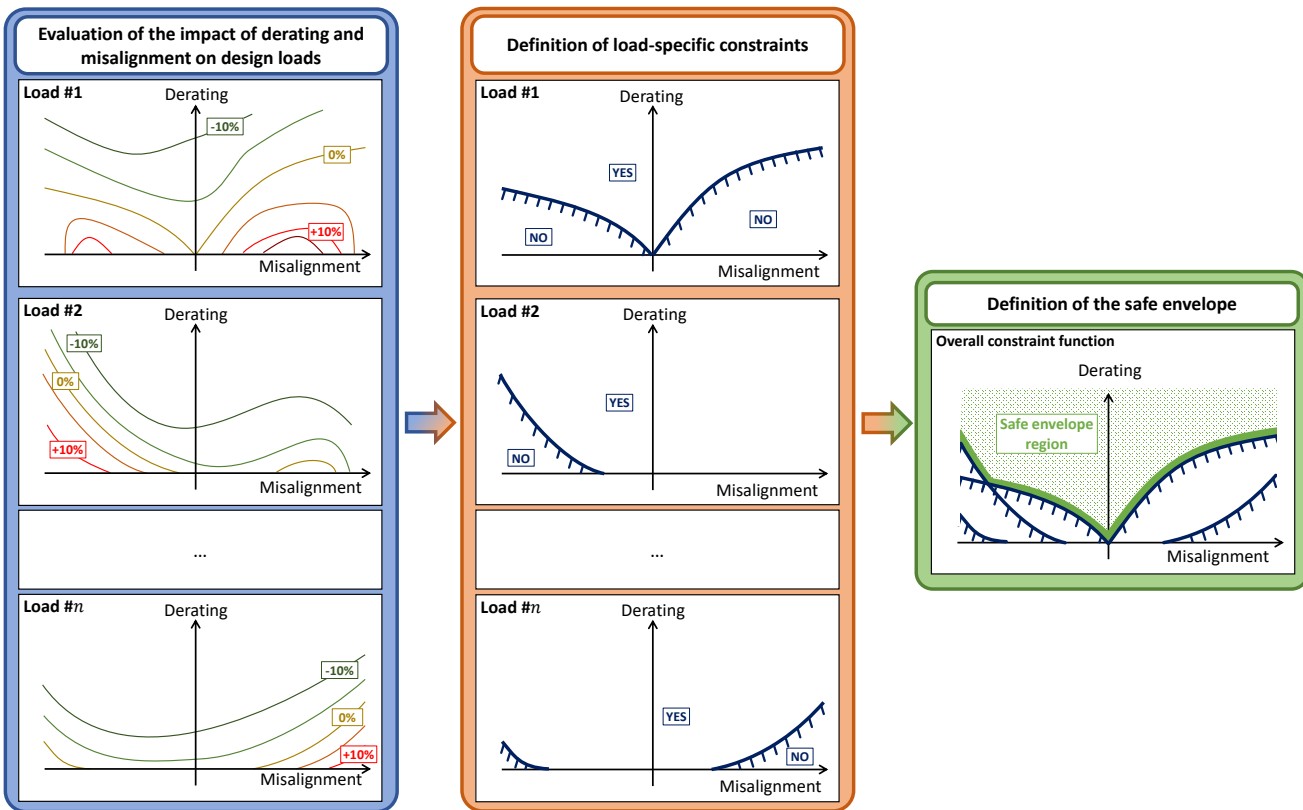

**Figure 1.** Sketch of the process for defining the safe envelope. Left (blue) block: qualitative contour plots of the increase in all design indicators (e.g. fatigue and ultimate loads or maximum displacements) in terms of misalignment and derating. Middle (orange) block: definition of the constraint function for each indicator, i.e. derating level that compensates for the possible increase in the specific load or displacement entailed by the misalignment. Right (green) block: combination of all load-specific constraints to define the **safe envelope region**.

The forthcoming sections 2.1 and 2.2, will better detailed the definition of the misalignment-derating constraint and the load-constrained maximization of the overall wind farm output respectively.





## 2.1 Definition of the misalignment-derating constraint through the analysis of fatigue and ultimate loads

As already discussed in the introduction, a parametric study of the impact of derated and yawed operations on ultimate and fatigue loads and maximum blade tip deflection of the isolated wind turbine is first analyzed. The behavior of downstream machines is not considered in this work and, clearly, this represents a limitation of the present analysis. The proposed approach is then to be viewed as a preliminary investigation on the possibility to combine wake redirection and derating farm control strategies to keep wind turbines operating within their load limits. The findings that will be detailed in Sec. 3 apply to the

sole turbines belonging to the front row of the farm but, clearly, an extension to downstream machines can be certainly done providing that a simulator of fatigue and ultimate indicators for all turbines of the farm is available.

The parametric analysis for the isolated turbine takes into account a subset of the well-known list of design load cases (DLC) (see IEC, 2005, Sect. 7.4), that are reported in Tab. 1. Such a shortlist was chosen according to the findings of Croce et al. (2022), which highlighted the most impacting cases in terms of rotor loads of the same turbine model considered in

this work, i.e. the INNWIND.EU reference 10-MW machine (Bak et al., 2013). Possible farm control failure modes are not considered within DLC2.x, because it is reasonable to think that supervisors of farm operations can be implemented so as to disengage the farm control if malfunctions are detected, in order to minimize their impact on the operations of the single turbines

**Table 1.** Definition of the DLCs considered in the analyses. NTM: normal turbulence model; ETM: extreme turbulence model; ECD: extreme coherent gust with direction change; EWS: extreme wind shear; EOG: extreme operating gust; EWM: extreme wind speed model.

| DLC | Wind Type | Wind speed | Horizontal Misalignment | Fault | Safety Factor | Performance indicator |
|-----|-----------|-----------|------------------------|-------|---------------|----------------------|
| 1.1/1.2 | NTM | $V_{in} : V_{out}$ | - | - | 1.35 | Fatigue and Ultimate |
| 1.3 | ETM | $V_{in} : V_{out}$ | - | - | 1.35 | Ultimate |
| 1.4 | ECD | $V_r, V_r \pm 2, V_{out}$ | - | | 1.35 | Ultimate |
| 2.2 PR | NTM | $V_{in} : V_{out}$ | - | Pitch Runaway | 1.1 | Ultimate |
| 2.3 | EOG | $V_r, V_{out}$ | - | Grid Loss | 1.1 | Ultimate |
| 6.2 | EWM | $V_{ref}$ | $-180 : 180$ deg | Grid Loss | 1.1 | Ultimate |
| 6.3 | EWM | $V_{ref}$ | $-20 : 20$ deg | - | 1.1 | Ultimate |

The DLC list should be repeated for different couples of turbine yaw misalignment angles and derating levels.

The misalignment angle $\phi$ corresponds to the angle between the wind direction and the projection of the rotor axis on a horizontal plane. A positive value of the misalignment angle is, in this work, associated with a counterclockwise rotation of the rotor seen from above. Consequently, the turbine experiences positive yaw misalignment if the wind velocity, viewed by an observer located on the rotor center and looking in front of the turbine, has a lateral component coming from the right side.



The derating $\xi$, on the other side, is defined as the ratio between the power reduction and the reference one in the same wind
condition. Accordingly,

$$P_{\text{der}}(V) = (1 - \xi) P_{\text{nom}}(V), \tag{1}$$

where $P_{\text{nom}}$ is the nominal power, $P_{\text{der}}$ is the derated one and $V$ is the wind velocity.

Fatigue and ultimate loads for all turbine sub-components are then stored in look-up tables generating an overall map related
to the variation of the design indicators as functions of the typical wind farm control parameters (i.e. yaw misalignment and
derating).

It is expected that independently of the turbine type and of its location within the farm derated operations will entail a
general reduction in maximum and fatigue loads. On the other side, yawed operations may increase the overall loading status
of the machine with respect to the reference condition associated with null yaw misalignment and null derating. The left block
in Fig. 1 provides a sketch of what one could expect from such a sensitivity analysis: all different design loads (fatigue and
ultimate loads or maximum displacement) feature a complex behavior (typically non-linear and discontinuous) in terms of
derating levels and misalignment angles, that can be quantified numerically.

Starting from this consideration, one can infer that given a specific load of interest, it is possible to find a derating level for
each yaw misalignment angle that compensates for the possible increase in the design loads.

To this end, consider the $j$th generic design indicator (e.g. maximum displacement, fatigue, or ultimate load) associated with
the $i$th turbine within the farm, named $y_{j_i}(\xi_i, \varphi_i)$ function of the turbine misalignment and derating. For all $y_{j_i}$ one can find
a compensating derating level by imposing the equality between the design indicator and those associated with the reference
condition, as

$$y_{j_i}(\xi_i, \varphi_i) - y_{j_i}(0, 0), \quad i = 1, \ldots, N_{\text{turb}}; \, j = 1, \ldots, N_{\text{ind}}; \tag{2}$$

where $\xi_i$ and $\varphi_i$ are respectively the derating and misalignment angle of the $i$th turbine, $N_{\text{turb}}$ is the number of the turbine
within the farm, $N_{\text{ind}}$ is the number of design indicators considered, while $y_{j_i}(0, 0)$ refers to the design indicator of the
reference conditions with $\xi_i = \varphi_i = 0$.

The function that solves Eq. (2) is named $f_{j_i}^{\text{cnstr}}$ and links the yaw misalignment angles with the load-compensating derating
$\xi_{j_i}^{\text{cnstr}}$ as

$$\xi_{j_i}^{\text{cnstr}} = f_{j_i}^{\text{cnstr}}(\phi_i). \tag{3}$$

The left and middle blocks of Fig. 1 qualitatively depict this process: from the map of the increase in the value of interested
design loads and displacements in terms of misalignment and derating, one can easily extract the constraint function $f_{j_i}^{\text{cnstr}}(\phi_i)$
seeking for the null increase. Notice that each load is expected to feature different constraint functions.

Once the load-specific constraints $f_{j_i}^{\text{cnstr}}$ of the $i$th turbine of the farm are determined for all indicators of interest, these
are to be combined to define the overall turbine constraint function $g_i^{\text{cnstr}}$, by selecting the most limiting constraints for each
misalignment as

$$g_i^{\text{cnstr}}(\phi_i) = \max_j \left( f_{j_i}^{\text{cnstr}}(\phi_i) \right). \tag{4}$$



The right block of Fig. 1 qualitatively shows how to combine the different load-specific constraints, into an overall one. Clearly, for each turbine in the farm, the constraint function $g_i^{\mathrm{cnstr}}$ splits the domain derating-misalignment into two regions. The first one, indicated with the green texture in Fig. 1 is the **safe envelope region** where all combinations of derating and yaw misalignment do not entail an increase in any of the design loads and displacements with respect to the reference conditions. Such a region can be mathematically indicated as

$$\xi_i \geq g_i^{\mathrm{cnstr}}(\phi_i). \tag{5}$$

It is expected that larger misalignment angles imply higher derating levels to compensate for possible increases in design indicators.

Then, a second region, when $\xi_i < g_i^{\mathrm{cnstr}}(\phi_i)$, comprises all combinations of misalignment and derating, where at least one load-specific constraint is violated, implying that it is possible that the $i$th turbine may experience an increase in that design load or displacement.

From a practical point of view, assuming that a turbine was designed and certified for standard operations with limited yaw misalignment angles, when it comes to implementing a farm control logic based on wake redirection, in order to keep the machine within its safe envelope, one has to simply enforce a certain level of derating when the turbine is subject to important yawed operations. Notice that this approach represents an interesting alternative to employing one-side wake steering policies or, in general, imposing strong limitations on the possible yaw misalignment angles, such as those used in many field applications of the wind farm control based on wake redirection (Fleming et al., 2019). In fact, it is not important to avoid a turbine working at misalignment angles potentially dangerous for its structural integrity, but rather to limit the increase in the machine loading status, a task that can be also accomplished by unloading the turbine through a suitable derating level.

Before describing possible uses of the safe envelope region within the synthesis of wind farm control, it is worthwhile clarifying some aspects.

First of all, the process described in this section to define the safe-envelope region could lead to a constraint curve excessively limiting. In fact, the DLC list, which is typically employed to design turbines, also considers extreme events and faults. Consequently, operating for a limited time outside the safe envelope does not necessarily imply a structural failure of the turbine. For example, if one expects that a too-large maximum tip deflection may be experienced at a large yaw misalignment during a coherent gust with extreme direction change (DLC 1.4), operating at that misalignment angle is not dangerous unless this extreme event happens. Clearly, such a risk is avoidable if the wind farm operator has one or more systems that allow the super-controller to measure and predict such extreme events in advance (e.g. a LIDAR). Moreover, the constraint curve, as it is defined in Eq. (5), applies to the turbine operations no matter the wind speed and turbulence. This fact could be limiting as the ultimate loads are typically experienced around the rated speed. Consequently, yawed operations at low speeds are seldom to be considered problematic. Clearly, a more thorough analysis could be performed so as to evaluate possible dependencies of the safe region with respect to wind velocity and turbulence intensity. This improvement of the process, although interesting, falls outside the scope of the paper, which is aimed at proposing a possible way of combining derating and wake redirection control in a load-constrained algorithm.



Being said this, the proposed approach, based on a combination of derating and misalignment, even without considering speed and TI dependency, results less restrictive than imposing strong limitations on the misalignment range such as those due to the one-sided wake steering, because the large and very large yaw angles, potentially dangerous, are not excluded but simply associated with a higher level of derating. It is expected that a more sophisticated definition of the safe envelope region (e.g.

including speed and TI dependency) may lead to improvement in the effectiveness of the proposed solution.

## 2.2 Load-constrained maximization of wind farm power output based on wake redirection and derating strategies

In this section, we will consider the definition of a wind farm control based on wake steering and derating, which includes also the load constraints as defined in Eq. (5).

Three different optimal setpoint definitions are considered. The first refers to the classical wake redirection control corre-

sponding to an unconstrained optimization of the misalignment angles of all turbines in the farm for maximum overall power production. The second strategy is a constrained optimization where both misalignment angles and derating levels are optimized, while the load-constraint in Eq. (5) for each turbine is enforced in the procedure. The third strategy represents a sub-optimal strategy that suitably mixes the previous approaches to limit the number of optimization parameters.

### 2.2.1 Reference wake redirection control

Consider a generic steady wind farm control based on the single wake redirection and unbounded, i.e. without considering the limitations entailed by structural issues. In this case, the control in its simplest definition consists in finding the yaw misalignment setpoints for each turbine belonging to the farm so as to maximize the produced power. To this end, let us collect the yaw $\phi$ of all turbines in an array $\mathbf{\Theta}^{\mathrm{ref}}$, as

$$\mathbf{\Theta}^{\mathrm{ref}} = \{\phi_1, \phi_2, \ldots, \phi_N\}, \tag{6}$$

where $N$ is the number of turbines in the farm, and on the ambient characteristics, i.e. wind speed $V$, turbulence intensity TI and direction $\phi_{\mathrm{wind}}$, in array $\mathbf{p}$, as,

$$\mathbf{p} = \{V, \mathrm{TI}, \phi_{\mathrm{wind}}\}, \tag{7}$$

The optimal control set-points related to the reference wake redirection strategy $\mathbf{\Theta}^{\mathrm{ref}}_{\mathrm{opt}}$ are computed by maximizing the overall farm power $P$ as

$$\mathbf{\Theta}^{\mathrm{ref}}_{\mathrm{opt}} = \arg\left(\max\left(P\left(\mathbf{\Theta}^{\mathrm{ref}}; \mathbf{p}\right)\right)\right), \tag{8}$$

where

$$P\left(\mathbf{\Theta}^{\mathrm{ref}}; \mathbf{p}\right) = \sum_i P_i\left(\mathbf{\Theta}^{\mathrm{ref}}; \mathbf{p}\right) \tag{9}$$

is the overall farm power, i.e. the sum of the power of the single turbines $P_i$ with $i = 1 : N$. Clearly, the power of the single turbines $P_i$ and, in turn, the overall farm power, depends on the yaw angles of all turbines $\mathbf{\Theta}$ and the ambient conditions $\mathbf{p}$.





Usually, problem (8) requires a non-linear optimization, typically gradient-based, with a number of optimization variables equal to the number of turbines belonging to the farm. For very large farms the optimization could be computationally expensive, especially if sophisticated simulation tools, such as those based on computation fluid dynamics, are used. For this reason, engineering or surrogate wind farm models can be employed to estimate the overall power as suggested by Doekemeijer et al. (2020) and Hulsman et al. (2020). Moreover, in order to limit the number of optimization parameters it is also possible to apply

the yaw control to a limited number of turbines, as proposed by Archer and Vasel-Be-Hagh (2019).

Obviously, since no bounds for the misalignment angles have been considered, it is possible that one or more turbines experience increases in their design loads and displacements.

### 2.2.2   Optimal load-constrained control combining wake redirection and derating

To cope with the possible increase in loads due to misaligned operations, it is proposed an optimal load-constrained control

that combines derating and wake redirection, exploiting the constraint function defined in Eq.(5).

To this end, the array of optimization variables is extended by adding the derating level of all turbines, as in $\boldsymbol{\Theta}^{\mathrm{comb}}$,

$$\boldsymbol{\Theta}^{\mathrm{comb}} = \{\phi_1, \phi_2, \ldots, \phi_N, \xi_1, \xi_2, \ldots, \xi_N\},  \tag{10}$$

where $\xi_i$ is the derating level of the $i$th turbine of the farm.

The load-constrained optimal combined control set-points for all machines are then computed by solving the following

constrained maximization problem,

$$\boldsymbol{\Theta}^{\mathrm{comb}}_{\mathrm{opt}} = \arg\left(\max\left(P\left(\boldsymbol{\Theta}^{\mathrm{comb}}; \boldsymbol{p}\right)\right)\right), \quad \mathrm{s.t.} \quad \xi_i \geq g_i^{\mathrm{cnstr}}(\phi_i), \quad i = 1, \ldots, N_{\mathrm{turb}}.  \tag{11}$$

The proposed methodology has some specific advantages with respect to other control techniques developed for a similar aim. Firstly, contrary to the method considered by Hulsman et al. (2020), it does not require the construction of a surrogate model from cost-expensive CFD simulations that would also result invariably dependent on the farm configuration and on

the selected scenarios employed to train the surrogate model itself. Secondarily, with respect to what was also proposed by Bossanyi (2018), we also include the ultimate loads and displacements in the analysis. In fact, the sizing of many turbine sub-components is driven by ultimate rather than fatigue loads and, as witnessed by Croce et al. (2022) and Bottasso et al. (2014), only the modification of such ultimate indicators has an effect on the design of the system.

### 2.2.3   Sub-optimal constrained and combined control

Clearly, the solution of problem (11) requires also the evaluation of the constraint for each wind turbine belonging to the farm, and its linearization in case gradient-based optimization techniques are employed. Moreover, the number of optimization parameters is doubled with respect to that of the unconstrained problem in Eq. (8).

In order to limit the number of optimization parameters, one can employ a sub-optimal algorithm that considers the load constraint in a practical and straightforward way starting from the setpoint obtained with the standard wake redirection control.





The idea comes from the fact that, as it will be demonstrated in Sec. 3.3, in terms of optimal misalignment angle $\phi_{\mathrm{opt}}$, the solutions of problems 8 and 11 are really similar for the majority of the analyzed cases. Along with a misalignment angle, the solution of the optimal combined problem features also a specific level of derating, which is needed to maintain the turbine operation within its safe envelope. From this consideration, one can easily devise a different setpoint definition that optimizes the sole misalignment angles without imposing the constraint, as it is done in the reference case object of Sec. 2.2.1.

Subsequently, the level of derating for each turbine is computed a posteriori as the minimum one that is needed to satisfy the constraint in (5). It is expected that this process will generate sub-optimal set-points compliant with the load constraint and associated with a limited reduction in farm power with respect to the optimal combined case.

To this end, one may define the array with the variable to-be-optimized in the sub-optimal case, $\mathbf{\Theta}^{\mathrm{constr}}$, exactly as it was defined in the reference wake redirection control

$$\mathbf{\Theta}^{\mathrm{constr}} = \mathbf{\Theta}^{\mathrm{ref}} = \{\phi_1, \phi_2, \ldots, \phi_N\}. \tag{12}$$

The optimal problem, as in the reference case, is formalized through an unconstrained optimization, as

$$\mathbf{\Theta}^{\mathrm{constr}}_{\mathrm{subopt}} = \arg\left(\max\left(P\left(\mathbf{\Theta}^{\mathrm{constr}}; \boldsymbol{p}\right)\right)\right), \tag{13}$$

that yields the sub-optimal set-points for all misalignment angles of all turbines.

Finally, the constraint is imposed for all turbines by finding the minimum derating level that satisfies Eq. (5). Such a derating

level can be computed as

$$\xi^{\mathrm{constr}}_{\mathrm{subopt}\,i} = g^{\mathrm{cnstr}}_i\left(\phi^{\mathrm{constr}}_{\mathrm{subopt}\,i}\right), \quad i = 1, \ldots, N_{\mathrm{turb}}, \tag{14}$$

where $\phi^{\mathrm{constr}}_{\mathrm{subopt}\,i}$ is the misalignment angle of the $i$th turbine computed through Eq. (13), while $\xi^{\mathrm{constr}}_{\mathrm{subopt}\,i}$ is the related derating level, evaluated to satisfy the load constraint. Notice also that the "greater-or-equal" symbol in Eq. (5) is now substituted by a simple "equal" sign, to emphasize that the derating is computed to bring the turbine operation on the onset of its safe envelope.

In order to show what we could typically expect from the three control techniques, a qualitative plot is depicted in Fig. 2. The behavior of the control methodology, hitherto expected, will be subsequently deeply analyzed in the result section (Sec. 3). The scheme refers to a hypothetical contour plot of the increase of the overall power related to a simple two-turbine farm for a generic wind condition. The horizontal and vertical axes are associated respectively with the misalignment angle and the derating level of the upstream machine, representing the sole optimization variables, since the downstream turbine will not

modify its operating condition. The contour is typically non-symmetric and may feature one or more local maxima. Moreover, too-high levels of derating may entail also a reduction in the overall power output. As wake redirection is typically more efficient in increasing overall farm output than steady derating, the absolute maximum is expected to be found in a condition with a specific misalignment and null derating. Consequently, we may envision that the solution of the reference wake redirection problem coincides with the absolute maximum as indicated by the square marker. Clearly, this behavior depends on

the specific case: as it will be shown in Sec. 3, it is possible to find the absolute maximum for non-null derating, especially when extremely reduced spacings are considered. On that plot, one may superimpose the load constraint, emphasizing what



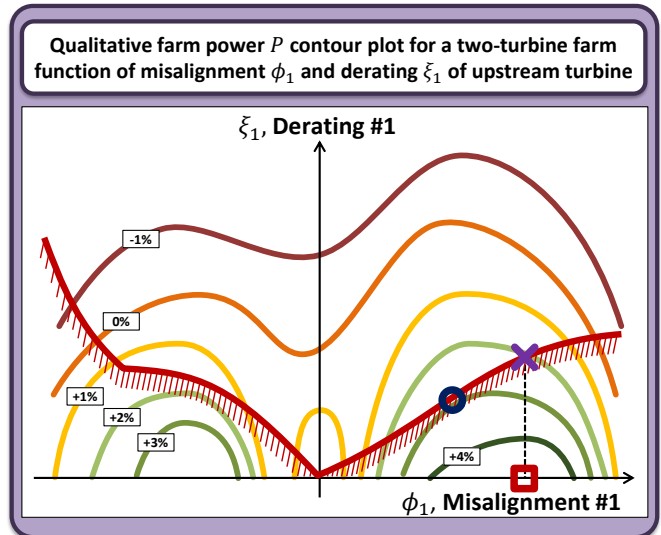

**Figure 2.** Qualitative definition of the control set-points for the three techniques in the case of a simple two-turbine farm, where only misalignment $\phi_1$ and derating $\xi_1$ of the upstream turbine are optimized. Contour plot: generic increase in the overall power as a function of $\phi_1$ and $\xi_1$. Solid red line: load constraint. Square marker: unconstrained optimum related to the reference wake redirection control. Circle marker: combined and constrained optimum. × marker: sub-optimum optimum.

we previously called "safe envelope region" (see Eq. 5). The lower bounds of the safe envelope region are reported as a solid red line. The solution of the load-constrained and combined control seeks the optimum in the upper part of the plot, where the constraint is satisfied. In general, imposing derating on the upstream machine, when it is yawed to deflect its wake out of downstream rotors, entails a reduction of the overall power output. This means that often, but not always, the optimal set-point of the combined and constrained control will be found exactly on the safe envelope boundary, where the constraint is tangent to the contour line with the highest power increase. This point is marked in the plot with a circle. Finally, the sub-optimal case is simply derived by projecting the solution of the reference wake redirection on the constraint, yielding the sub-optimal set-point in correspondence with the × marker.

As it will be demonstrated later on, the difference between the solution of the combined and sub-optimal control is typically marginal, as well as the performance of the two techniques in terms of overall farm power increase. Hence, one could in principle define the operative setpoints of the turbines in a sub-optimal through a simpler optimization algorithm without a strong detrimental impact on the overall power output.



## 3 Results

### 3.1 Definition and modeling of the reference wind turbine and farm


The study object of this work considers the 10 MW INNWIND.EU model. This turbine is an upwind pitch-regulated machine of a diameter equal to about 178 m, which is described in Bak et al. (2013). The model was implemented in the general-purpose multibody software `Cp-Lambda` (Bottasso and Croce, 2009–2018; Bottasso et al., 2006). Turbine flexible elements, such as the blades, the tower, and the shaft, are implemented through a nonlinear, kinematically exact beam formulation

with fully-populate cross-sectional stiffness matrices (Bauchau, 2011). Rotor aerodynamics is rendered through the classical blade element momentum theory. Lifting lines are used for modeling blade aerodynamic forces and moments, as well as for reproducing the drag of the nacelle and the tower. Hub- and tip-losses and tower shadow are also considered. The overall modeling of the aerodynamics implemented in `Cp-Lambda` was also recently validated against field data coming from a 2.3 MW wind turbine with a diameter of 80 meters in sheared and yawed inflow conditions (Boorsma et al., 2023).

The turbine model considers also a first-order dynamic model for the generator and a second-order one for pitch actuators.

The control of the turbine in operating conditions is managed by the POLI-Wind controller and is based on a linear quadratic regulator (LQR), as described in Riboldi (2012) and Bottasso et al. (2012). Such an LQR controller is based on the linearization of the torque balance equation about the trimming conditions, a task that is easy to be performed providing the availability of the standard $C_p - \lambda$ curves. The LQR, being model-based, results to be extremely flexible in managing also the derating. In

fact, by linearizing the model about different reference conditions, that are derived so as to generate a desired derating level, one automatically obtains the new control gains, without modifying the architecture of the control.

In particular, in derated conditions, the operating point of the turbine was found by modifying only the pitch settings leaving unaltered the tip-speed ratio with respect to the nominal case at the same wind speed. Hence, for all wind speeds from the cut-in to cut-out, the pitch setting, realizing the desired derating level $\beta_{\mathrm{derat}}$, was found by solving the following equation

$$\frac{1}{2}\rho V^3 S C_p\left(\lambda_{\mathrm{nom}}, \beta\right) = (1 - \xi) P_{\mathrm{nom}}(V), \tag{15}$$

where $\rho$ is the air density, $\lambda_{\mathrm{nom}}$ is tip-speed-ratio associated with the nominal case, $\beta$ the pitch settings and $C_p\left(\lambda, \beta\right)$ are the well-known $C_p - \lambda$ look-up table. Given $\beta_{\mathrm{derat}}$ and $\lambda_{\mathrm{nom}}$ for all desired derating levels, it is also possible to evaluate the thrust coefficients, by simply interpolating the thrust coefficient look-up table, $C_t\left(\lambda, \beta\right)$.

Finally, misalignment angles are reproduced in the simulation by rotating the nacelle. Control architecture and gains are not

modified in yawed conditions with respect to the reference, i.e. aligned, case.

Idling, faults, startup and shutdown maneuvers, and, more in general, all the non-operating conditions and the transition among the different operating states, are managed by the POLI-Wind Supervisor (Riboldi, 2012).

The analyzed wind farms are modeled through the software `FLORIS` (FLOw Redirection and Induction in Steady State), jointly developed by NREL and TUDelft (NREL, 2019). This software features a multitude of engineering wake models, which

can be employed to characterize the steady flow within the farm. The turbines are included in the farm according to their power and thrust coefficients defined as functions of the wind speed. The outputs of all turbines and, in turn, the overall farm power



production are computed according to the mean flow at each rotor location. In this work, the Gaussian wake model and wake combination based on kinetic energy were employed.

Within FLORIS, the impact of misalignment on wake deflection is included in the wake models, whereas derating can be
rendered by modifying the power and thrust coefficients of each turbine. Dynamical variations of wind direction, derating, and misalignment are not considered.

### 3.2 Definition of the load constraint and of the safe envelope region

The DLC list in Tab. 1 was simulated with the reference 10 MW turbine model so as to evaluate fatigue and ultimate loads as well as maximum blade tip deflections for all combinations of five yaw misalignment angles $(0, \pm15, \pm25)\,\mathrm{deg}$ and five
derating levels $(0, 2.5\%, 5.0\%, 10\%, 15\%)$.

Clearly, the DLC6.n series was considered only for the nominal case with null misalignment and null derating, as it refers to a non-operative phase where any farm control is not active. Since this case can result in design loads, however, it is critical to take this into account in this scenario as well. This, in general, raises another general issue regarding the impact of a wind farm controller on ultimate loads or other indicators: if design loads came from not-controlled conditions (e.g. parking) or from
conditions where the wind farm control does not operate (at high winds, for example), then clearly this constitutes a condition that can allow the controller to operate without any constraints.

In this analysis, only wind farm controls scheduled with respect to the sole wind speed are considered, and an upper limit for its activation equal to 15 m/s is imposed, no matter the turbulence intensity (TI), as it was already done by Croce et al. (2022).

Finally, notice that the list in Tab. 1 refers to a selection from the whole cases prescribed by the Standards. In fact, only the
most impacting DLCs were considered on the basis of previous analysis on the same machine, reported in detail in Croce et al. (2022).

in order to present the idea, in this paper only rotor loads are taken into account, neglecting the impact of wake redirection and derating on tower and hub loads. However, the very same process can be extended to other turbine sub-components, without any modifications.
Figure 3 shows the two most affecting ultimate indicators related to the blade. The blade-root combined ultimate load and the maximum blade tip deflection are displayed respectively on the left and on the right. Both plots present the percentage variation of such indicators as functions of the misalignment and the derating. In order to evaluate the maximum blade tip deflection, we considered the displacements of all three blades within an azimuthal segment of 60 deg around the tower position, so as to capture only those conditions exposed to the risk of blade-tower strike.
Qualitatively, both plots feature similar trends: significant increases are experienced at higher misalignment angles. Furthermore, as expected, even small derating levels entail sensible reductions in the indicators, such that it is even possible to fully compensate for the increments caused by large misalignment angles. Notice also how the plots are non-symmetric as the maximum increases in blade root combined loads are experienced for negative misalignment angles, while the blade tip deflection is highest for negative ones. Quantitatively, at -25 deg the derating level that compensates for the increase in blade root loads





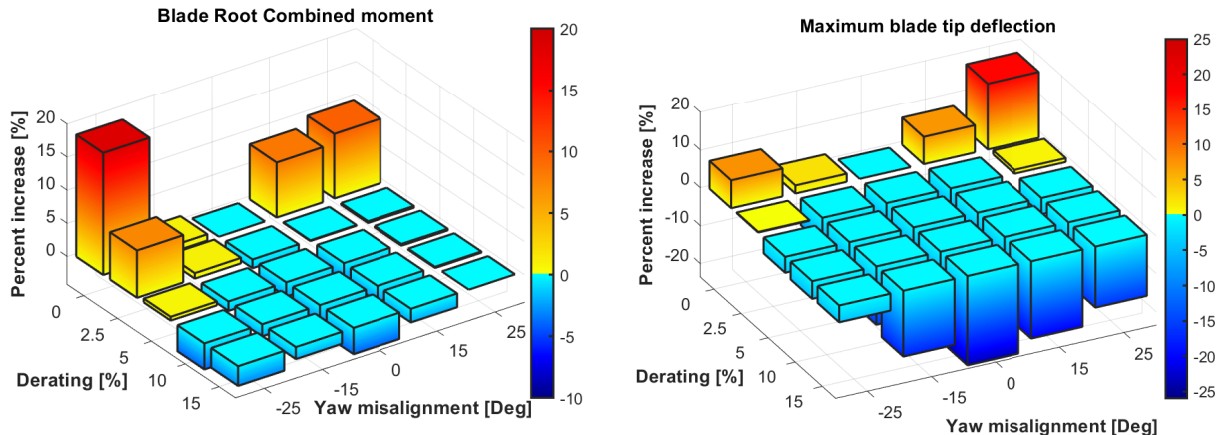

**Figure 3.** Percentage variation of blade root combined load (left) and maximum blade tip deflection (right) as functions of misalignment and derating, taken with respect to nominal condition (i.e. null misalignment and derating)

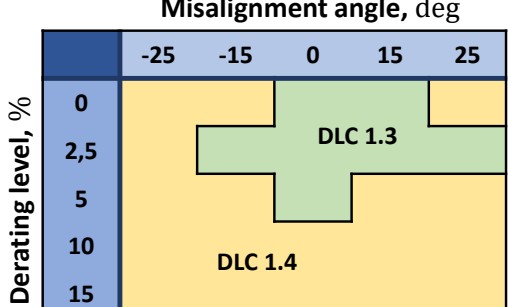

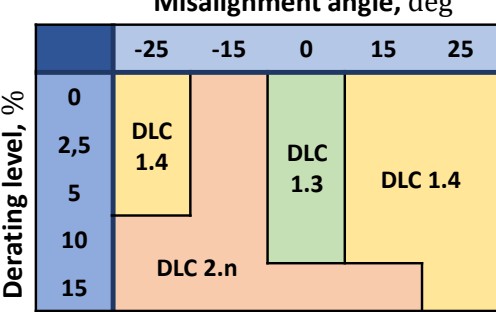

**Figure 4.** DLC types associated with the ultimate indicators for blade root combined load (left) and maximum blade tip deflection (right) as functions of misalignment and derating.

is slightly higher than 5%, whereas at +25 deg the increase in maximum blade deflection is compensated by a derating slightly less than 5%. Such values are apparently limited and compatible with turbine standard operating conditions.

Figure 4 shows the DLC type reporting the ultimate indicators for blade root combined load (left) and maximum blade tip deflection (right). Notice how the cases associated with ultimate loads and displacements may vary on the basis of the chosen parameters. This represents further evidence that a global overview of all operative regimes of the turbines is needed to evaluate the actual impact of the farm-controlled operations on the design drivers.



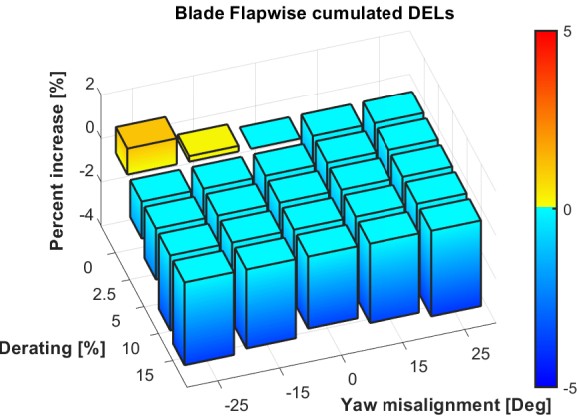

**Figure 5.** Percentage increase in blade root flapwise fatigue load as a function of misalignment and derating, taken with respect to nominal condition (i.e. null misalignment and derating)

Fatigue loads, on the other side, seem less impacted by the misalignment, as can be noticed in Fig. 5 showing the cumulated DEL for blade root flapwise. The maximum percentage increase is limited, i.e. less than 2%, and is experienced only for positive and large misalignment angles. Here again, as in the previous analyses, it can be clearly noticed the positive impact of derating on loads.

From the maps of the variation of the design indicators, one can easily find the specific constraint functions defined in Eq. (3) by numerically solving Eq. (2). Dealing with the solution of Eq. (2), the maps associated with loads and displacements were linearly interpolated.

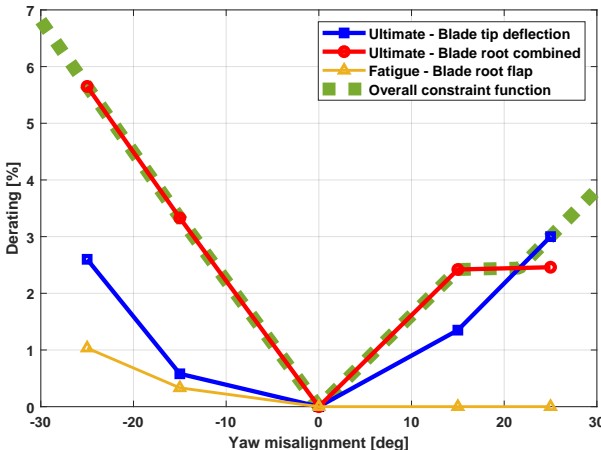

**Figure 6.** Constraint function definition for blade root combined ultimate load, maximum blade tip deflection, and blade root flapwise fatigue load.





Figure 6 displays the constraint functions, i.e. the derating level that compensates for the possible increase in loads and displacements induced by misalignment, for the three analyzed quantities (square markers: blade root combined ultimate load;

circle markers: maximum blade tip deflection; triangle marker: blade root flapwise fatigue load).

As it can be easily seen from the graph, for this machine, the most constraining function is associated with the blade root combined ultimate load, while the fatigue of the blade root flapwise features an inactive constraint, as it is associated with lower derating levels for all misalignment angles. The constraint associated with the maximum blade tip deflection, on the other side, is active only for the highest positive misalignment angles.

Finally, the overall constraint function, visualized by a dashed thick line, can be simply extracted from load-specific constraints by taking the maximum derating level as defined in Eq. (4).

To cover the range of misalignment angles outside the bounds of the performed analyses, i.e. $(\pm 25)\deg$, the overall constraint function was linearly extrapolated.

### 3.3 Optimal load-constrained farm control for different inflow conditions, spacing, and overlaps

In this section, the performance of the reference, combined, and sub-optimal controllers is evaluated in terms of the overall power output of a simple two-turbine wind farm. The analyses were made for different inflow conditions, defined in terms of wind speed $V$ and turbulence intensity TI, and different geometries of the farm, defined in terms of the spacing $s$ between the turbines and the lateral offset $y_h$ of the downstream machine. Figure 7 offers an easy-to-interpret visual representation of the farm geometry parameters. In particular, we considered all combinations of the following parameters,

$$\text{Wind speed}: V = (7, 10, 11.4, 12, 12.5, 13, 14)\,\text{m/s},$$
$$\text{Turbulence intensity}: \text{TI} = (2\%, 6\%, 10\%),$$
$$\text{Spacing}: s = (3D, 4D, 5D, 6D, 7D),$$
$$\text{Lateral Offset}: y_h = (\pm 1D, \pm 0.75D, \pm 0.5D, \pm 0.25D, 0); \tag{16}$$


for a total number of 945 cases.

The optimal load-constrained solutions of Eq. (11) and of the sub-optimals one of Eq. (13) presented here below, have been computed in MATLAB, by solving the constrained optimization problem with a gradient based algorithm (i.e. *fmincon*) coupled with the wind farm solver `Floris` (NREL, 2019).

At first, consider a reference case with wind velocity $V = 14\,\text{m/s}$, corresponding to the rated speed, TI $= 6\%$, spacing $s = 5D$, being $D$ the rotor diameter, and lateral spacing $y_h = 0$ representing a full wake impingement.

Figure 8 shows the contour plot of the farm power gain as a function of the upstream misalignment and derating. The overall constraint function, a solid red curve, is superimposed on the contour, splitting the domain into two regions, where the constraint is satisfied (i.e. the safe envelope) and where it is not. Moreover, Tab. 2 summarizes the control set-points and the

associated power gains for the three control strategies.

The standard unconstrained wake redirection control finds the optimum at about $24.6\,\deg$ and obviously with null derating, outside the safe envelope, with a net increase in the farm output of $5.4\%$. The optimum found according to the combined

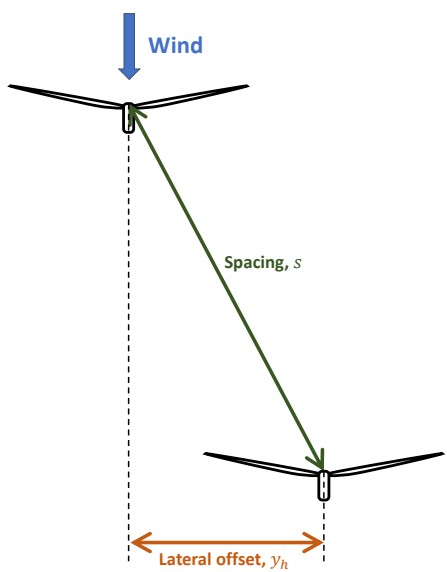

**Figure 7.** Top view of the simple two-turbine farm with the most important geometry parameters, i.e. spacing $s$ and lateral offset $y_h$. The lateral offset is positive if the downstream turbine is on the right with respect to wind direction if viewed from above.

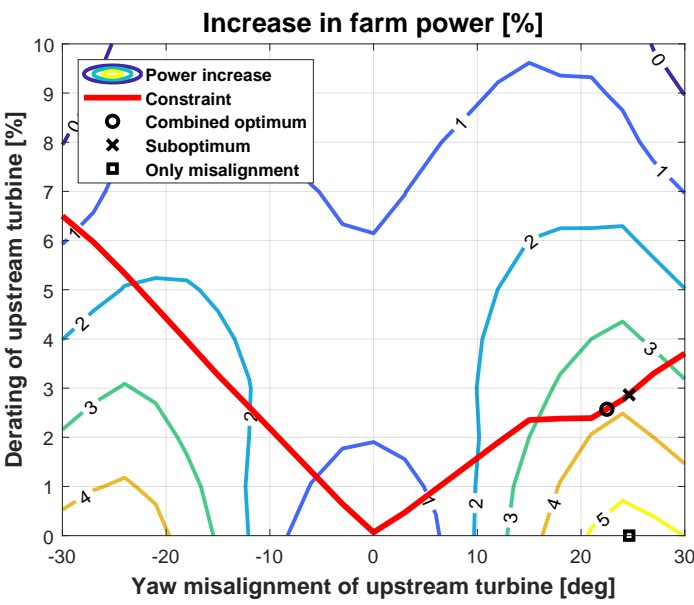

**Figure 8.** Contour plot of the overall power output of the two-turbine farm with $s = 5D$, $y_h = 0$ and at $V = 11.4 \, \mathrm{m/s}$ and $\mathrm{TI} = 6\%$. Red solid line represents the overall constraint function. Optimal control setpoints are also visualized as a square marker (reference wake redirection, only misalignment), circle marker (combined and constrained control) and × marker (sub-optimal control).



**Table 2.** Optimal setpoint and power gain. Two-turbine wind farm with $s = 5D$ and $y_h = 0$, at $V = 11.4\,\text{m/s}$ and TI = 6%.

|  | Misalignment | Derating | Power increase |
|---|---|---|---|
|  | deg | % | % |
| Combined | 22.5 | 2.6 | 3.9 |
| Suboptimum | 24.6 | 2.9 | 3.8 |
| Reference WR | 24.6 | 0.0 | 5.4 |

strategy, on the other hand, features a slightly lower misalignment (i.e. $24.6\,\text{deg}$) and a mild level of derating (i.e. $2.6\%$). Notice that this set-point lays exactly on the constraint line. As expected, the inclusion of the constraint into the optimization strategy entails a reduction of the power gain of about $1.5$ percentage points. In any case, the gain associated with the combined strategy, $3.9\%$, still remains significant. Consequently, derating the upstream machine, while it is yawed, to fulfill the load constraint, reduces but does not annihilate the effectiveness of the wake redirection.

An important discussion should be made concerning the sub-optimal approach. As said in Sec. 2.2.3, combined and reference wake redirection strategies typically feature similar optimal misalignment angles. This fact is clearly visible in the results shown in Fig. 8 and Tab. 2, providing preliminary evidence of the rationale behind the sub-optimal control.

The analyzed case shows that, in the sub-optimal control, the derating set-point of the upstream machine (i.e. $2.9\%$) is higher than that of the combined strategy as a result of the higher optimal misalignment angle. The gain increment in the sub-optimal control $3.8\%$ , however, results to be marginally lower than the one associated with the combined one equal to $3.8\%$. The fact that the two controls are characterized by similar performance could favor the use of the simpler strategy, the sub-optimal one, especially in larger farms when the turbines to be controlled are much more.

**Table 3.** Optimal set-point and power gain. Two-turbine wind farm with $s = 5D$ and $y_h = 0.5D$, at $V = 10.0\,\text{m/s}$ and TI = 6%.

|  | Misalignment | Derating | Power increase |
|---|---|---|---|
|  | deg | % | % |
| Optimal combined | 18.6 | 2.38 | 7.53 |
| Suboptimal combined | 19.1 | 2.39 | 7.52 |
| Reference WR | 19.1 | 0.0 | 8.72 |

Tables 3 and 4 report the optimal parameters and gains for two other cases. The first one refers to a condition with wind speed $V = 10.0\,\text{m/s}$ and TI = 6% with spacing equal to $5D$ and lateral offset equal to $0.5D$ corresponding to a partial wake impingement on the left side of the downstream rotor. The second case is characterized by wind speed $V = 11.4\,\text{m/s}$ and TI = 6% with spacing equal to $7D$ and lateral offset equal to $-0.5D$, corresponding now to an impingement on the right side of the downstream machine.

From both tables, one can immediately notice that the same conclusions, that were derived for the case related to the full impingement scenario (see Fig. 8 and Tab. 2), are still valid. In particular, the optimal misalignment for the combined control





**Table 4.** Optimal setpoint and power gain. Two-turbine wind farm with $s = 7D$ and $y_h = -0.5D$, at $V = 11.4\,\mathrm{m/s}$ and TI $= 6\%$.

|  | Misalignment | Derating | Power increase |
|---|---|---|---|
|  | deg | % | % |
| Optimal combined | -15.2 | 3.3 | 6.3 |
| Suboptimal combined | -17.2 | 3.7 | 6.1 |
| Reference WR | -17.2 | 0.0 | 7.9 |

is slightly lower in modulus than the one related to the reference wake redirection control and features a significant power gain even if lower than the one related to the unconstrained case. The difference in power gain is about $1.5$ percentage points.

On the other side, sub-optimal control performs similarly to the combined one, a marginally lower power gains, i.e. less than $0.2$ percentage points. This last remark demonstrates again that sub-optimal control represents a valuable strategy.

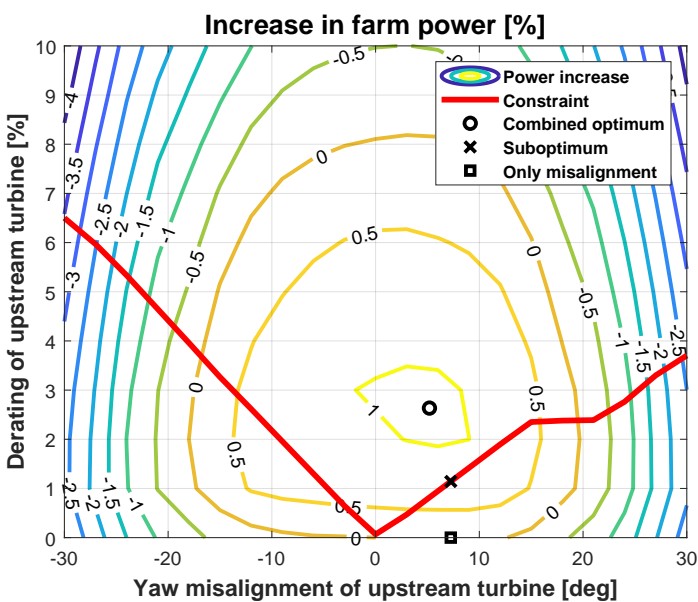

**Figure 9.** Contour plot of the overall power output of the two-turbine farm with $s = 3D$, $y_h = 0$ and at $V = 11.4\,\mathrm{m/s}$ and TI $= 6\%$. The red solid line represents the overall constraint function. Optimal control set-points are also visualized as a square marker (reference wake redirection, only misalignment), circle marker (combined and constrained control), and $\times$ marker (sub-optimal control).

A last case was also considered to show how the three controllers might behave differently in extreme cases, such as those related to very tight spacing. To this end, Fig. 9 and Tab. 5 reports the results of an analysis with a spacing $s = 3D$, lateral offset $y_h = 0$, at $V = 11.4\,\mathrm{m/s}$ and TI $= 6\%$.

In this case, the pure wake redirection is not as effective as in the previous conditions as the close spacing between the turbine does not allow the wake to actually move away from the downstream rotor. Derating strategy, on the other hand, appears to be



**Table 5.** Optimal set-point and power gain. Two-turbine wind farm with $s = 3D$ and $y_h = 0$, at $V = 11.4\,\mathrm{m/s}$ and TI $= 6\%$.

|  | Misalignment | Derating | Power increase |
|---|---|---|---|
|  | deg | % | % |
| Optimal combined | 4.6 | 2.6 | 1.1 |
| Suboptimal combined | 7.2 | 1.2 | 0.8 |
| Reference WR | 7.2 | 0.0 | 0.1 |

more effective. In this extreme situation, the optimal combined control outperforms the standard wake redirection and features optimal misalignment angles noticeably lower, i.e from $7.2\,\mathrm{deg}$ to $4.6\,\mathrm{deg}$. As a consequence of that, the differences in the performance between combined and sub-optimal controls are quite evident.

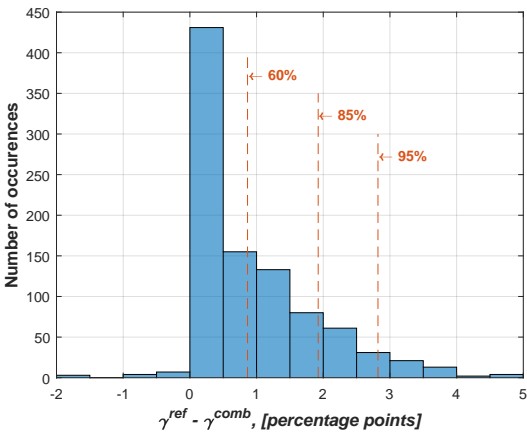
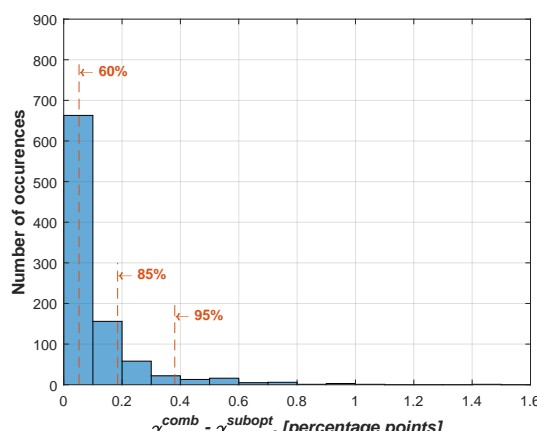

**Figure 10.** Histograms representing the number of occurrences associated with the differences between the power gains associated with the different strategies. Left plot: difference between the percentage gain associated with reference wake redirection control $\gamma^{\mathrm{ref}}$ and that related to the combined one $\gamma^{\mathrm{ref}}$. Right plot: difference between the percentage gain associated with the optimal combined control $\gamma^{\mathrm{comb}}$ and that related to the sub-optimal one $\gamma^{\mathrm{subopt}}$. Vertical dashed lines indicate the range associated with the $60\%$, $85\%$, and $95\%$ of the occurrences.

An overall analysis of the differences in the performance of the three controls, considering the whole set of 945 analyzed conditions, was also performed with the aim of analyzing this procedure applied to this specific wind farm in more detail. Figure 10 displays the histograms representing the number of occurrences associated with specific differences in terms of power gain between the reference wake redirection and the optimal combined controls, on the left, and between combined and sub-optimal controls, on the right. The number of occurrences is reported in the $y$-axis while the difference in the power gains
is in the $x$-axis.

From the left plot of Fig. 10, it is possible to verify that the overall impact of the load constraint on the power output is in general limited, as the difference between the percentage gain associated with reference control $\gamma^{\mathrm{ref}}$ and that related to





the combined one $\gamma^{\mathrm{ref}}$ is quite limited for the majority of the analyzed cases. As indicated by the vertical dashed lines, the difference between the gains of the two strategies is below $0.8$ percentage points for the $60\%$ of the cases, below $2$ percentage

points for the $85\%$ of the cases, and below $2.75$ percentage points for the $95\%$ of the cases. Negative values in the $x$-axis of the left plot in Fig. 10 refer to those extreme conditions with spacing equal to $3D$, for which the optimal combined control outperforms the reference one.

The difference between combined and sub-optimal controls is even less marked, as witnessed by the right plot of Fig. 10. For the $95\%$ of the cases, the power gain of the sub-optimal control is lower than that of the combined one for at most $0.4$

percentage points.

### 3.4   Evaluation of control performance for a nine-turbine wind farm.

In order to evaluate and compare, in a more complex scenario, the behavior of the controls, described in Sec. 2.2, a nine-turbine farm is considered.

The farm layout is organized such that the farm is square-shaped with three rows and three columns spaced by $5\,D$. The farm

is also oriented so as to have the diagonals aligned in the North-South and East-West directions, respectively. The direction the wind is blowing from is indicated with $\phi_{\mathrm{wind}}$. When $\phi_{\mathrm{wind}} = 0\,\mathrm{deg}$ the wind is blowing from the North, whereas when $\phi_{\mathrm{wind}} = 90\,\mathrm{deg}$ it is blowing from the East.

The wind farm layout is represented in Fig. 11, where also the turbine numbering is reported.

Two relevant conditions are considered. The first considers and inflow with speed $V = 9\,\mathrm{m/s}$ and direction $\phi_{\mathrm{wind}} = 0\,\mathrm{deg}$,

i.e. coming from the North. The second one refers to the same speed but to a different direction, $\phi_{\mathrm{wind}} = 90\,\mathrm{deg}$, representing the most impacting condition, being associated with minimum distance between upstream and downstream rotors and with full impingement levels.

In this analysis, it is assumed that the constraint function $g_i^{\mathrm{constr}}(\phi_i)$ stays the same for all turbines, independently of the location within the farm, as

$$g_\ell^{\mathrm{constr}}(\phi_\ell) = g_\kappa^{\mathrm{constr}}(\phi_\kappa), \quad \forall\,(\ell;\kappa) \in (1, \ldots, N_{\mathrm{turb}}; 1, \ldots, N_{\mathrm{turb}}) \qquad (17)$$

Tables 6 and 7 display the results of the two analyses, reporting, along with the turbine-specific and overall power gains, the derating and the misalignment angle of all machines, for all controls.

From the results obtained considering the nine-turbine wind farm, one can derive the same conclusions related to the simpler cases, analyzed in Sec. 3.3. In particular, the optimal combined control features lower optimal misalignment in the upstream

turbines with respect to those associated with the reference control. The inclusion of the load constraint entails a mild reduction in the optimal farm output, without, in any case, jeopardizing the effectiveness of the wake redirection strategy. Finally, with respect to the optimal combined control, the sub-optimal one performs similarly, with a negligible reduction in the gain potentially achievable.





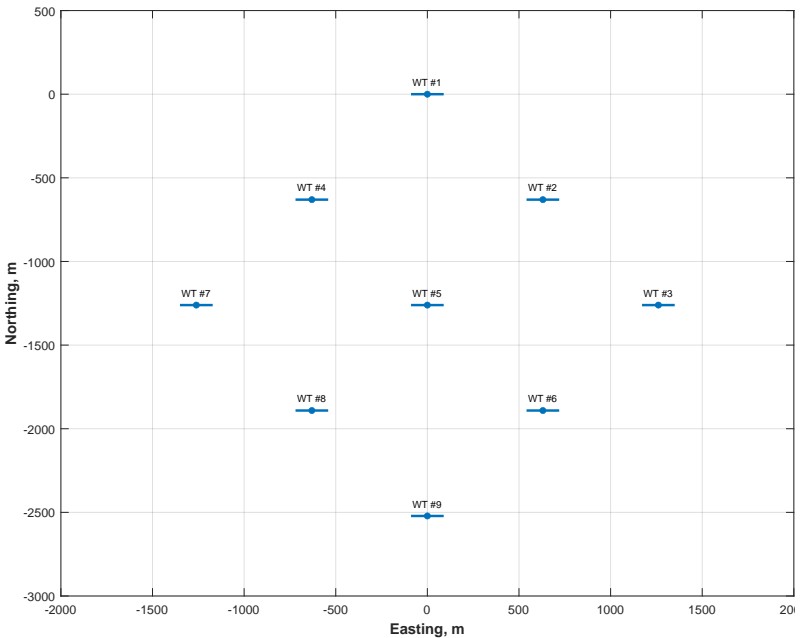

**Figure 11.** Generic nine-turbine wind farm layout

**Table 6.** Wind farm analysis at $V = 9\,\mathrm{m/s}$ and $\phi_{\mathrm{wind}} = 0\,\mathrm{deg}$. $\phi_i$ and $\xi_i$: misalignment angle in $\mathrm{deg}$ and derating level of the $i$ turbine; $\gamma$: overall farm power gain in percentage. The behavior of WT#4, WT#7, and WT#8 is respectively equal to that of WT#2, WT#3 and WT#6. The performance of WT#8 is equal to that of WT#6.

|  | WT#1 | | WT#2 | | WT#3 | | WT#5 | | WT#6 | | WT#9 | | Overall power gain |
|---|---|---|---|---|---|---|---|---|---|---|---|---|---|
|  | $\phi_i$ | $\xi_i$ | $\phi_i$ | $\xi_i$ | $\phi_i$ | $\xi_i$ | $\phi_i$ | $\xi_i$ | $\phi_i$ | $\xi_i$ | $\phi_i$ | $\xi_i$ | $\gamma$ |
| Optimal combined | 22.9 | 2.6 | 21.5 | 2.4 | 0 | 0 | 19.4 | 2.4 | 0 | 0 | 0 | 0 | 4.4 |
| Suboptimal combined | 25.1 | 2.9 | 22.9 | 2.6 | 0 | 0 | 19.3 | 2.4 | 0 | 0 | 0 | 0 | 4.3 |
| Reference WR | 25.1 | 0 | 22.9 | 0 | 0 | 0 | 19.3 | 0 | 0 | 0 | 0 | 0 | 5.7 |

## 4 Conclusions and outlook

In this paper, we first presented a parametric analysis related to the impact of the combination of derating and misalignment angles on the design indicators (i.e. fatigue and ultimate loads, and maximum displacements) of an isolated multi-MW wind turbine. The parametric analysis, thanks to the fact that derating entails a general reduction in the loading status of the turbine, was then used for deriving a safe envelope region, i.e. a region where the combination of the two parameters is not associated with an increase in any of the machine design indicators. The safe envelope can be easily formulated in terms of the minimum

derating level needed to compensate for possible increases in design indicators due to yawed operations. From the performed




**Table 7.** Wind farm analysis at $V = 9\,\mathrm{m/s}$ and $\phi_{\mathrm{wind}} = 45\,\mathrm{deg}$. $\phi_i$ and $\xi_i$: misalignment angle in $\deg$ and derating level of the $i$ turbine; $\gamma$: overall farm power gain in percentage. The behavior of WT#2 and WT#3 is equal to that of WT#1. The behavior of WT#5 and WT#6 is equal to that of WT#4. The behavior of WT#8 and WT#9 is equal to that of WT#7.

|  | WT#1 | | WT#4 | | WT#7 | | Overall power gain |
|---|---|---|---|---|---|---|---|
|  | $\phi_i$ | $\xi_i$ | $\gamma_i$ | $\phi_i$ | $\xi_i$ | $\gamma_i$ | $\gamma$ |
| Optimal combined | 25.5 | 3.1 | 23.8 | 3.0 | 0 | 0 | 14.7 |
| Suboptimal combined | 28.4 | 3.5 | 24.6 | 2.9 | 0 | 0 | 14.7 |
| Reference WR | 28.4 | 0 | 24.6 | 0 | 0 | 0 | 17.8 |

analyses, such compensating derating was expressed as a nonlinear function of the sole misalignment angle, neglecting possible dependencies on wind speed and turbulence intensity.

Finally, two novel wind farm control techniques, based on a combination of wake redirection and derating, are proposed. The main idea is to exploit a combined use of derating and misalignment to optimize farm production while maintaining all
the turbines in the farm within the respective save envelope region.

In a combined approach, the turbine operation set-points, in terms of derating and misalignment, are defined so as to maximize the farm power subject to the constraint that the operation of all turbines be inside the safe envelope. In a suboptimal approach, the load constraint is directly imposed on the optimal misalignment angles defined by the standard wake redirection control.

The entire procedure, from the computation of the safe envelope region to the definition of the load-constrained controls, was tested in a simulation environment using a 10MW reference turbine model and two simple reference farms, made by two and nine turbines.

From the results shown in this paper, as well as from extensive practice on the same topic, the following conclusions can be derived.

– Derating the turbine when it is yawed has the advantage of reducing the impact that misaligned operations may have on turbine design indicators. In particular, not only blade fatigue loads are reduced but also ultimate loads and maximum blade tip deflections.

– With a simple parametric study, it is possible to find a derating level, function of the misalignment angles, that is able to compensate for the possible increase in the design indicators entailed by the yawed operations. In this study, the
compensating derating level resulted to be quite small, of the order of a few percentage points.

– When tested on simple two-turbine and nine-turbine farms the optimal combined control performs as expected. Clearly, a penalty of a few percentage points in the power gain is experienced as a result of the imposition of the load constraint. However, the optimal combined farm control remains highly effective in increasing the overall power output.



– With extremely low spacing, i.e. $3D$, the optimal combined control outperforms the standard wake redirection. This was due to the fact that for such reduced spacing the derating is more effective than the wake redirection technique.

– The sub-optimal control, interestingly, features a performance in terms of overall power gain, really similar to that of the optimal combined control, and requires an optimization with fewer variables.

– With the proposed controls it is no more necessary to limit the yaw misalignment angle of the upstream turbine or to employ "one-sided wake steering" methodology to limit the impact on loads due to large yaw operations. The "derating vs misalignment" constraint automatically ensures that the turbine operates within its load limits.

Clearly, the work object of this paper represents a preliminary investigation and much is to be done before this technique may reach maturity.

Firstly, the definition of the safe envelope region should include a dependency on wind speed and, possibly, turbulence intensity. This may improve the effectiveness of the optimal combined control, that, as it is implemented in this work, considers the load constraint also at low speeds, which are typically associated with a minor impact on the loading status of the machine.

Secondarily, it will be important to include in the formulation also fatigue and ultimate loads of the downstream machines. In fact, the constraint function was here computed only for the upstream turbine in out-of-wake conditions. This will be investigated in further research activities.

Last, the proposed methodology, with a few modification in the tools used, may be used to further analize the combination of other wind farm control techniques (i.e. dynamic induction controls)

Finally, the inclusion of the proposed controls in a turbine design framework is certainly an extension of interest.

All the aforementioned extensions and possibilities are currently under investigation.

*Data availability.* The data used for generating all figures will be made available for download through a data sharing platform, e.g. Zenodo, in the final revised version of this paper.

*Code and data availability.* The linked Matlab-Python tool implementing the combined controls can be made available upon request.

*Author contributions.* All authors provided fundamental inputs to this work through discussions, feedback, and analyses of the obtained results. AC and SC devised the main idea of the load-constrained controls and wrote the manuscript, AC supervised the research activities. FI performed the Floris analyses.

*Competing interests.* AC is member of the editorial board of *Wind Energy Science*.





*Acknowledgements.*  The Authors gratefully acknowledge Gianluca Dadda, M.Sc., for his contribution to the preparation of the multibody aero-servo-elastic simulations of the design load cases for the reference 10MW wind turbine combining wake redirection and derating.



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
