# Peer review of "Combining wake redirection and derating strategies in a wind farm load-constrained power maximization"

_Wind Energy Science, 2023_

## Referee Comment (RC1)

**Reviewer comments to wes-2023-145 "Combining wake redirection and derating strategies in a wind farm load-constrained power maximization" by A. Croce et al.**

The submitted paper deals with the integration of loads into wind farm control. The subject is very relevant. Industry has been hesitant to introduce wind farm control because it is unclear how it affects loads and fatigue damage. Usually, control algorithms maximize only power production. The authors propose to introduce loads into the search for higher AEP through constraining the optimization function. The contraints are based on design load cases and include ultimate and fatigue loads. Two control actions are combined. Wake steering through misaligning the turbine and turbine derating. While wake steering generally increases loads, derating decreases them. Through this, a strategie where the turbine operates in a safe envelope can be found. The paper is well written and structured. It is recommended for acceptance after minor revisions.

Comments:

- The authors do not discuss the wake behaviour due two the possible combination of derating and wake steering. Are there any studies that investigate the combination of derating with wake steering? How does the derating affect the wake displacement?

- Adding to this point, it is also not deeply discussed how these phenomena are modelled in FLORIS. To the knowledge of the reviewer, there are currently no explicit models for derating in the wake models in FLORIS. How does the derating affect the wake added turbulence and wake recovery? Are there hints that the modelling through changing the Ct curve of the turbine is sufficient? How does derating affect a steered wake in FLORIS?

- Line 408: It says twice 3.8%

- Figure 11 has a very small font

- It seems that turbine 5 in figure 11 is both waked and performing control actions. Does the impinging wake of turbine #1 influence the result? It is hinted before, that the method so far is only applicable for free stream turbines. If that is the case, it is a bit confusing to show also an optimized turbine #5.

---

## Author Comment (AC1)

Dear Editor, Referee #1,

First and foremost, thank you for reviewing our paper "Combining wake redirection and derating strategies in a wind farm load-constrained power maximization" (Preprint wes-2023-145). We are glad that your feedback was positive.

We have revised the original manuscript to accommodate what you suggested. We have also taken this opportunity to make minor improvements to the text.

We believe that the manuscript has improved, and we hope that this amended version will satisfy your expectation for this work.

The point-by-point reply to your comments is reported here below.

*[Reviewer] The authors do not discuss the wake behaviour due two the possible combination of derating and wake steering. Are there any studies that investigate the combination of derating with wake steering? How does the derating affect the wake displacement?*
*Adding to this point, it is also not deeply discussed how these phenomena are modelled in FLORIS. To the knowledge of the reviewer, there are currently no explicit models for derating in the wake models in FLORIS. How does the derating affect the wake added turbulence and wake recovery? Are there hints that the modelling through changing the Ct curve of the turbine is sufficient? How does derating affect a steered wake in FLORIS?*
[Answer] The derating of the turbine indirectly affects the wake displacement through the implied variation of the thrust coefficient $C_T$. In fact, a derated turbine works at a lower power coefficient and different thrust coefficient than the nominal ones. We did not develop a dedicated model for wake behavior in yawed and derated operations, but we simply based this research activity on the modeling already implemented in FLORIS, which follows the treatment of Bastankhah and Porté-Agel (Journal of Fluid Mechanics. 2016;806:506-541. doi:10.1017/jfm.2016.595). According to this model, the turbine thrust coefficient affects multiple wake characteristics including the onset of the far-wake region, TI, speed deficit and lateral displacement of the wake center. Bastankhan and Porté-Agel model is considered adequate, at least for the scope of our work, to capture the combined impact of derating and misalignment on wake behavior.
[Action] We added a short paragraph to better explain this. This paragraph was added at the end of section 3.1.

*[Reviewer] Line 408: It says twice 3.8%*
[Answer] The reviewer is right. The second value should be 3.9% as reported in the table.
[Action] The text was modified.

*[Reviewer] Figure 11 has a very small font*
[Answer] We agree with the Reviewer.
[Action] The font size of the text in the figure was increased and the figure itself enlarged.

*[Reviewer] It seems that turbine 5 in figure 11 is both waked and performing control actions. Does the impinging wake of turbine #1 influence the result? It is hinted before, that the method so far is only applicable for free stream turbines. If that is the case, it is a bit confusing to show also an optimized turbine #5.*
[Answer] Yes, clearly, the internal turbines belonging to a row of three or more machines can be waked and perform control actions, as the Reviewer pointed out. Floris framework can model this situation and suitably compute the power output of internal turbines. The assumption that we made in this analysis is that the possible presence of the wake does not significantly influence the load-constraint functions. This means that the load constraints can be evaluated once for the isolated turbine and then applied to all turbines belonging to the farm. Probably, this assumption is strong for fatigue loads but may be acceptable for ultimate ones, that, in the case at hand, represent the active constraints.

In section 3.4, we wrote "In this analysis, it is assumed that the constraint function $g_i^{\mathrm{constr}}(\phi_i)$ stays the same for all turbines, independently of the location within the farm", to emphasize this.
Moreover, the same concept is expressed in the conclusion ("Secondarily, it will be important to include in the formulation also fatigue and ultimate loads of the downstream machines. In fact, …")
[Action] The sentence of section 3.4 was extended a bit to stress the limitation of the proposed approach.

We look forward to your kind reply, and in the meanwhile we send our warmest regards.

Sincerely yours,

Alessandro Croce and Stefano Cacciola

---

## Author Comment (AC2)

Dear Editor, dear Daan van der Hoek

First and foremost, thank you for reviewing our paper "Combining wake redirection and derating strategies in a wind farm load-constrained power maximization" (Preprint wes-2023-145). We are glad that your feedback was positive.

We have revised the original manuscript to accommodate what you suggested. We have also taken this opportunity to make minor improvements to the text.

We believe that the manuscript has improved, and we hope that this amended version will satisfy your expectation for this work.

The point-by-point reply to your comment is reported here below.

*[Reviewer]* The paper contains many instances where one-sentence paragraphs could easily be combined with the preceding or following paragraph.
[Action] We have carefully read the manuscript and combined some sentences as suggested by the reviewer. At the same time, we took this opportunity to make minor improvements to the text.

*[Reviewer]* Ln 13: Power maximization and cost minimization are interconnected and should not be regarded as two separate aspects.
[Answer] We partially agree with the Reviewer. Surely, power and cost are interconnected, but, in the design and control of wind energy systems, maximizing power and minimizing cost may lead to different design choices. This basically happens because power is mainly driven by the aero-servo-elastic characteristics, while costs depend also on many other quantities, including price and quantity of the used materials, maintenance, actuator duty cycle, terrain lease and so on.
[Action] Even if in the present paper we considered only power maximization, the sentence, as it is now, is correct and does not need to be amended.

*[Reviewer]* Ln 22: Please add a reference supporting this statement.
[Answer] We fully agree with the Reviewer's suggestion.
[Action] We added here references to Damiani et al (2018) and Croce et al. (2022).

*[Reviewer]* Ln 32: Please specify whether this concerns the yawed turbine, downstream turbines, or both.
[Answer] The work performed in the specific references (Boorsma (2012), Damiani et al. (2018) and Croce et al. (2022)), considered only the yawed and isolated turbine.
[Action] The sentence was slightly modified to accommodate Reviewer's suggestion.

*[Reviewer]* Ln 45: Please specify how derating is achieved in this paper.
[Answer] We understand the point raised by the Reviewer, however, since there are multiple ways for derating a turbine, we prefer to be generic at this point, as the mentioned arguments apply to all possible derating techniques. The precise procedure we employed to derate the turbine is reported in Section 3.1 ("In particular, in derated conditions, the operating point of the turbine was found by modifying only the pitch settings leaving unaltered the tip-speed ratio…")
[Action] Here again, we do not see the need for modifying the text in this section.

*[Reviewer]* Ln 182-185: Similarly, wake redirection is only employed for a limited range of wind directions. This could also be considered during the load-constrained optimization, as the safe region corresponds to average fatigue loads, correct?
[Answer] This is correct. In the safe envelope definition, we considered the fatigue loads that are computed assuming all operating speeds (from cut-in to cut-out) weighted with the Weibull distribution. In principle,

since low speeds mildly contribute to fatigue, one could accept an increase in DEL for those conditions. This possibility is not considered in the present work but represents a possible extension of the proposed strategy. [Action] We wrote "…with respect to wind speed, direction and turbulence intensity".

*[Reviewer]* Ln 220: Can you be more specific on how the optimization is limited to a smaller number of turbines? For example, by making the yaw misalignments of downstream turbines a function of the upstream turbines.
[Answer] In the paper of Archer and Vasel-Be-Hagh (2019), they simply yawed the turbines that mostly affect the overall power production.
[Action] The text was modified to better explain this fact.

*[Reviewer]* Ln 279-281: Not only will derating the turbine result in lower power output, but the decrease in thrust will also make wake steering less effective. Please also comment on this aspect of the combined control approach.
[Answer] This is correct and was similarly raised by the other reviewer.
[Action] We commented on this aspect at the end of section "Definition and modeling of the reference wind turbine and farm", as this discussion fits perfectly here.

*[Reviewer]* Section 3.1 does not present any results and would fit better in the methodology section. This would also prevent some questions that arise after reading the current section 2 on how the combined control strategy is implemented.
[Answer] We agree with the reviewer.
[Action] Subsection 3.1 was moved outside Section Results and promoted as an independent section.

*[Reviewer]* Ln 323: Which wake deflection model is employed within FLORIS?
[Answer] The wake deficit, expantion and deflection are model according to the Gaussian Model described in Bastankhah and Porté-Agel, *Journal of Fluid Mechanics*, 2016 and Niayifar and Porté-Agel, *Energies*, 2016.
[Action] The last part of the section "Definition and modeling of the reference wind turbine and farm" was enlarged to better explain this.

*[Reviewer]* Figure 3: While the 3d bar plots clearly show the results of some of the edge cases, it is not possible to distinguish the effects of yaw misalignment and derating in the center of the plots. Please switch to a 2D contour plot, or 2D bar plot, or make the current bars uniform in color (no gradient). The same applies to Fig. 5.
[Answer] Thank you again for the suggestion. We had tried during the writing of the paper and tried again now to use different graphs to represent these load changes. Since, however, we think it is useful to show the two effects (yaw and derating) at the same time, we believe this picture is more meaningful. Moreover, the exact values to exactly reproduce these figures will be provided with the final version of the paper.
[Action] For the reasons explained above we believe that it is better to leave the figures as they are and provide all the values to replicate the plot through a data sharing platform.

*[Reviewer]* Figure 6: Is this constraint function independent of wind speed? Please specify if this is not the case.
[Answer] Yes, the constraint function is independent of the wind speed. This assumption was done to simplify the treatment as already explained at the end of section 2.1.
[Action] A sentence was added to the text commenting figure 6, to stress this fact.

*[Reviewer]* Ln 390: Rated speed of the 10MW turbine is set at V=11.4 m/s and not 14 m/s.
[Answer] The Reviewer is right.
[Action] We correct the typo 14 in 11.4.

*[Reviewer]* 16: When considering the test cases, we see that most wind speeds are above-rated. What is the reasoning behind this parameter choice, given that wake redirection is most effective at below-rated conditions (if at all effective above-rated)?

[Answer] Below 7 m/s the effect of the control is limited due to low wind speeds, while above 14 m/s the farm control is ineffective (not necessary) because low thrust coefficients, characterizing the upstream turbine, entail limited speed deficit inside the wake. Hence, we focused on the speed region where we do expect the highest impact of the farm control and where possible discontinuities are present due to the different operating conditions (i.e. below and above rated speed).

[Action] We added a sentence to explain it better immediately after equation (6). We also emphasized that a wider speed range is necessary for a comprehensive design and evaluation of the impact of the farm control.

*[Reviewer]* Figure 7: Would it not make more sense to refer to spacing s as the vertical between the turbines, and not the absolute distance between turbines as is used at the moment?

[Answer] The way we defined the geometry parameters is kinematically correct to represent the situation in which the turbines are fixed, and the wind can rotate. In any case, a simple computation shows that the difference in the two definitions is about 6% in the worst one of the analyzed cases, that is spacing of 3D and lateral offset of 1D.

[Action] No modification to the text is required.

*[Reviewer]* Figure 11: Please increase font size. Furthermore, I suggest changing the axes to express distance as a function of rotor diameter D.

[Answer] We agree with the Reviewer.

[Action] The figure was increased in font sizes and the axes were modified as suggested.

*[Reviewer]* Ln 455: The second case should refer to 45 deg instead of 90 deg.

[Answer] The reviewer is right.

[Action] We changed also this typo as requested.

*[Reviewer]* While the results with the 9 turbine farm demonstrate the effectiveness of the proposed control methods, the symmetry of the layout results in some redundant results (as also indicated in the table captions). Therefore, it would be interesting to consider a more complex/less uniform layout.

[Answer] We agree with the Reviewer that the considered case is really simple. However, the goal of that section is to provide an example of the feasibility and robustness of the proposed approach when one has multiple optimization parameters. In fact, doubling the number of variables while introducing nonlinear constraints may in principle render the optimization problem unaffordable, or, at least, less robust. The fact the obtained solutions reflect the symmetry of the farm layout is, from this point of view, a good indicator.

[Action] We modify the first sentence of the section to clarify the actual scope of the example.

*[Reviewer]* The constrained optimization currently only considers blade root moments and blade tip deflection. Are these also the most critical components to consider, or will including other structural components further decrease the performance gain of wake redirection with derating?

[Answer] Yes, it is possible that the inclusion of other loads (e.g. tower loads or hub loads) may lead to a narrower safe envelope. This should be highlighted.

[Action] A new sentence has been added to the conclusion to highlight this point.

*[Reviewer]* While the choice of a steady-state wake model to demonstrate the proposed combined wind farm controller is understandable, these models are prone to overestimating the effects of derating on the wake statistics. Please comment on this in the paper in the discussion of the simulation results with 3D distance.

[Answer] Reviewer is right. More in general, a tuning of the wake model parameters based on higher-fidelity simulation or, better, field data is necessary, even if the obtained results do not differ too much from previously published material related to field or wind tunnel testing of axial induction control (see

Bossanyi&Riusi *Wind Energy Science* 2021, Campagnolo et al. *IFAC* 2023, Van der Hoek et al. *Renewable Energy* 2019)

[Action] A new sentence, along with additional new references, has been added at the point where we discussed the simulation results at 3D distance. Another sentence was added to the conclusions to emphasize the point raised by the Reviewer.

*Technical corrections*

[Reviewer] Title: "load-constrained wind farm power maximization"
[Answer] We agree.
[Action] We modified the title as suggested.

[Reviewer]
Ln 10: "… while maintaining an unaltered design load envelope …"
Ln 20: "Among all …, wake redirection (WR) has proved to be highly effective for increasing wind farm energy harvesting."
Ln 21: "… deflect its wake away from downstream rotors."
Ln 23-24: Single sentence paragraph, combine with subsequent paragraph.
Ln 27: incomplete reference "Services (2004)".
Ln 36: "misaligned" instead of "misalignment".
Ln 45: "… integrated in load-constrained wind farm control."
Ln 60: "increased design loads".
Ln 74: Two consecutive sentences starting with "moreover".
Ln 94: Rewriting the sentence is suggested, for example: "Clearly, to keep every turbine operating below or at its design loads, a wind farm controller should maximize power output within the safe envelope of each turbine in a farm, thereby combining derating and wake redirection."
Ln 182: "… seldom considered problematic."
Ln 186: "That being said, …"
Ln 186-189: For clarity, please rewrite into multiple smaller sentences.
Ln 193: " … a wind farm controller …"
Ln 200: "steady-state wind farm controller"
[Answer] We thank the Reviewer for all these comments and suggestions.
[Action] We modified the text as suggested.

[Reviewer] Ln 201: "… consists of …"
[Answer] The form "… consists in doing something" should be correct.
[Action] We did not implement any action.

[Reviewer]
Ln 202: "… yaw angle \phi …"
Ln 205: Break in two sentences. "Similarly, the ambient characteristics are collected into an array p …"
Ln 224: "…, an optimal load-constrained controller is proposed …"
Ln 251: "… in Eq. (5)."
Answer] We thank the Reviewer for all these comments and suggestions.
[Action] We modified the text as suggested.

[Reviewer] Ln 276: "On that plot, …". I assume this is referring to Fig. 2, but please specify which plot.
Answer] We thank the Reviewer for pointing out this unclear sentence.

[Action] We modified the text in this way: "Superimposed to the contour plot of the farm power increase, as in Fig. 2, one can draw the load constraint, emphasizing what we previously called ``safe envelope region'' (see Eq. 5).".

*[Reviewer]*
*Ln 287: "… define the turbines' operative setpoints through a simpler sub-optimal optimization algorithm without a …"*
*Ln 304: "The LQR, being model-based, …" Please rewrite the sentence for clarity.*
*Ln 339: Missing reference for "Standards".*
*Ln 342: Capital letter "in order …"*
[Answer] We thank the Reviewer for all these comments and suggestions.
[Action] We modified the text as suggested.

*[Reviewer] Ln 354: I believe there is a typo here, blade tip deflection is highest for positive misalignment angles (Fig. 3b).*
[Answer] The text should be correct as it is. The increase in blade root loads is higher at -25 deg (fig 3a) and needs a compensating derating slightly higher that 5%. Whereas the increase in blade tip deflection is higher at 25 deg and needs slightly less 5% of derating (Fig 3b).
[Action] We modified the text adding the reference to the left and right plot of Fig. 3 to improve clarity.

*[Reviewer]*
*Ln 363: "As in the previous analyses, the positive impact of derating on loads can be clearly noticed."*
*Ln 391: "…, with D being the rotor diameter, …"*
*Ln 407: "However, the gain increment in the sub-optimal case (i.e. 3.8%), appears to be only marginally lower than that of the combined control approach equal to 3.9%."*
*Ln 410: "…, especially in farms containing a large number of turbines."*
*Ln 423: "To this end, Fig. 9 and Tab. 5 report …"*
*Ln 439: "… for 60% of the cases, … for 85% of the cases, … for 95% of the cases."*
*Ln 514: typo "analyze"*
[Answer] We thank the Reviewer for all these comments and suggestions.
[Action] We modified the text as suggested.

We look forward to your kind reply, and in the meanwhile we send our warmest regards.

Sincerely yours,

Alessandro Croce and Stefano Cacciola